# Immunoassay System Based on the Technology of Time-Resolved Fluorescence Resonance Energy Transfer

**DOI:** 10.3390/s24051430

**Published:** 2024-02-22

**Authors:** Zhengping Xu, Hong Zhou, Li Li, Zhang Chen, Xin Zhang, Yongtong Feng, Jianping Wang, Yuan Li, Yanfan Wu

**Affiliations:** 1Suzhou Institute of Biomedical Engineering and Technology, Chinese Academy of Sciences, Suzhou 215163, China; zhouh@sibet.ac.cn (H.Z.); lil@sibet.ac.cn (L.L.); chenz@sibet.ac.cn (Z.C.); zhangxin@sibet.ac.cn (X.Z.); fengyt@sibet.ac.cn (Y.F.); wangjianping@sibet.ac.cn (J.W.); wuyf@sibet.ac.cn (Y.W.); 2Chongqing Guoke Medical Innovation Technology Development Co., Ltd., Chongqing 401122, China; liyuan@sibet.ac.cn

**Keywords:** time resolved, fluorescence resonance energy transfer, immunoassay, in vitro diagnosis

## Abstract

To enhance the specificity and sensitivity, cut the cost, and realize joint detection of multiple indicators, an immunoassay system based on the technology of time-resolved fluorescence resonance energy transfer (TR-FRET) was studied. Due to the FRET of the reagent, the donor probe and acceptor probe emitted specific fluorescence to enhance specificity. Long-lifetime specific fluorescence from the acceptor probe was combined with time-resolved technology to enhance sensitivity. A xenon flash lamp and a photomultiplier tube (PMT) were selected as the light source and detector, respectively. A filter-switching mechanism was placed in the light path, so the fluorescence signal from the donor and acceptor was measured alternately. The instrument’s design is given, and some specificI parts are described in detail. Key technical specifications of the instrument and procalcitonin (PCT), C-reactive protein (CRP), and interleukin-6(IL-6) were tested, and the test results were presented subsequently. The CV value of the self-designed counting module is better than 0.01%, and the instrument noises for 620 nm and 665 nm are 41.44 and 10.59, respectively. When set at 37 °C, the temperature bias (B) is 0.06 °C, and the temperature fluctuation is 0.10 °C. The CV and bias are between ±3% and 5%, respectively, when pipetting volumes are between 10 μL and 100 μL. Within the concentration range of 0.01 nM to 10 nM, the luminescence values exhibit linear regression correlation coefficients greater than 0.999. For PCT detection, when the concentration ranges from 0.02 ng/mL to 50 ng/mL, the correlation coefficient of linear fitting exceeds 0.999, and the limit of quantification is 0.096 ng/mL. For CRP and IL-6, the detection concentration ranges from 0 ng/mL to 500 ng/mL and 0 ng/mL to 20 ng/mL, respectively, with limits of quantification of 2.70 ng/mL and 2.82 ng/mL, respectively. The experimental results confirm the feasibility of the technical and instrumental solutions.

## 1. Introduction

In vitro diagnostics (IVD) technology is an indispensable component of modern medical practice, involving the analysis of human samples such as blood, body fluids, and tissues outside the body to provide critical clinical diagnostic information [1,2]. Molecular diagnostics, immunoassays, and biochemical tests represent the three major domains of IVD, with immunoassays holding a significant market share in China [3,4]. Among these, magnetic particle chemiluminescence immunoassay technology is widely employed in the field of immunoassays due to its advantages of short detection time and high precision [5,6,7]. However, this technique requires a high level of operational expertise, and the performance of the magnetic beads and the magnetic separation process can affect the accuracy and reliability of the results. Furthermore, it is based on the biotin–avidin mechanism [8]. While biotin is an essential vitamin for the human body [9], oral intake may lead to false-positive results, and the presence of anti-avidin antibodies within the human body can result in inaccurate detections.

To address these challenges, time-resolved fluorescence resonance energy transfer (TR-FRET) technology [10,11,12,13] has emerged as a promising solution. TR-FRET [14] is based on the principle of fluorescence resonance energy transfer (FRET) [15,16,17], which achieves specific detection of target molecules through the energy transfer between two fluorescent labeling molecules within a certain distance [17]. This approach is further enhanced by incorporating time-resolved measurement techniques [18,19], wherein a delay time is set to bypass the short-lifetime background fluorescence at the moment of light emission, effectively distinguishing the target signal from background noise [20]. This significantly enhances the sensitivity and accuracy of detection [21]. Another major advantage of this technique is its no-wash requirement, which simplifies the testing process, reducing both operational complexity and costs.

Although high-performance instruments based on TR-FRET technology, such as the B•R•A•H•M•S KRYPTOR Compact PLUS by Thermo Fisher Scientific, are already available on the market, the prohibitive costs and the restrictions associated with proprietary reagents remain significant barriers to their widespread adoption. This situation underscores the urgent need for the development of a TR-FRET immunoassay system that is both cost-effective and maintains superior performance.

This study aims to design and develop a cost-effective immunoassay system based on TR-FRET technology, focusing on the multiplexed detection of novel inflammatory biomarkers [22,23], including Procalcitonin (PCT) [24,25,26], C-Reactive Protein (CRP) [27], Serum Amyloid A (SAA), and Interleukin-6 (IL-6). These biomarkers play crucial roles in the early diagnosis and treatment of inflammation-related diseases. We aspire to provide an economical and user-friendly solution, offering a more accurate and reliable tool for the early diagnosis and monitoring of inflammatory diseases.

In summary, this study not only responds to the urgent clinical demand for high-performance, low-cost in vitro diagnostic (IVD) technologies but also holds the promise of positively impacting public health by advancing immunodiagnostic technology, thereby enhancing the efficiency and effectiveness of disease management.

## 2. Principle of TR-FRET Immunoassay

The TR-FRET immunoassay method combines the FRET technology of the reagent with the time-resolved measuring technology of the instrument [28,29]. As an example, using the sandwich method, the schematic diagram of fluorescence resonance energy transfer is shown in Figure 1.

The core of FRET are two fluorophores: the donor and the acceptor. When exposed to the excitation light, the donor emits its own fluorescence. When the conditions for FRET are met, the donor transfers a portion of its energy to the acceptor through FRET before emitting fluorescence. This results in the emission of fluorescence at a specific wavelength by the acceptor. The conditions for FRET include appropriate alignment of the dipoles of the donor and acceptor, overlapping fluorescence spectra of the donor and excitation spectra of the acceptor, and a distance between the donor and acceptor within the range of 1 to 10 nm. The efficiency of FRET is inversely proportional to the sixth power of the distance between the donor and acceptor [30,31]. Therefore, the distance between them significantly influences the intensity of fluorescence emitted by the acceptor.

Typically, the fluorescence lifetime of organic dyes is within the nanosecond range, and the Stokes shift, which is the difference between the maximum excitation and emission wavelengths, spans several tens of nanometers. In contrast, lanthanide ions [32] exhibit much longer fluorescence lifetimes. For example, the fluorescence lifetime of Eu (Europium) chelates can reach the millisecond range, with a Stokes shift of approximately 300 nm. Therefore, Europium (Eu^3+^) cryptate [33] is selected as the donor probe with its strongest fluorescence emission wavelength at 620 nm; correspondingly, Cy5 is chosen as the acceptor probe, which has its strongest fluorescence emission wavelength at 665 nm. Under non-FRET conditions, the acceptor probe emits short-lifetime fluorescence. When FRET conditions are met, the donor probe can continuously transfer energy to the acceptor probe within its fluorescence lifetime, leading the acceptor probe to emit long-lifetime specific fluorescence [34]. The principle of time-resolved fluorescence resonance energy transfer (TR-FRET) measurement is illustrated in Figure 2. It is important to note that the term ‘long-lifetime’ is relative to the microsecond-level short-lifetime of background fluorescence, typically being in the millisecond range.

As shown in Figure 2, the black curve represents background fluorescence, and *t*_1_ indicates its lifetime, which is usually in microseconds. The brown curve represents short-lifetime fluorescence from the acceptor with non-FRET. The short lifetime is close to that of background fluorescence, so measurement results with high SNR could not be gained. As mentioned above, the lifetime of the donor probe is long, as shown by the green curve in Figure 2. With FRET, the donor probe transfers energy to the acceptor probe, so the lifetime is extended, as shown by the red curve in Figure 2.

To avoid the influence of short-lifetime background fluorescence, the time-resolved measurement [35,36,37] is induced, as illustrated in Figure 2. The triggering time of the light source is recorded as zero time. Instead of starting measurement immediately upon excitation, a delayed time is induced. When the delayed time has passed, the intensity of background fluorescence becomes very low. At the same time, signal acquisition starts, so a high SNR could be gained. Fluorescence signal intensity from the donor probe and the acceptor probe were labeled separately as SD and SA, and the ratio of SA to SD was used as the measuring result to determine the concentration of the analyte.

In summary, the core principle of time-resolved measurement lies in the selection of specific time intervals of interest. FRET underpins time-resolved measurement by producing long-lifetime fluorescence, which is instrumental in distinguishing signal from short-lifetime background fluorescence.

## 3. Measuring Module

The optical system is the heart of the measuring module. The principal diagram of the optical system of the measuring module is depicted in Figure 3.

Considering the desired quantitative limits and the instantaneous power of the light source, we have selected the Hamamatsu L11947 xenon flash lamp as the system’s light source in Beijing. Referring to the L11947 datasheet, its emission spectrum ranges from 185 nm to 2500 nm, with a power output of 20 W and a lifespan of up to 1 × 10^8^ flashes.

The light from the xenon flash lamp passes through a lens for collimation before entering a narrow-band filter1. The center wavelength of the narrow-band filter1 is determined by the absorption spectrum of the donor probe. As mentioned above, an Eu acupoint-like fluorescent chelator is chosen as the donor probe, so the center wavelength of filter1 is selected as 320 nm. The excitation light is reflected by a dichroic mirror and then focused on the sample cup using a lens. The reflection band of the dichroic mirror should ensure that light from filter1 can be reflected. In the sample cup, two reagents and samples form immune complexes. When excited, the immune complexes emit fluorescence, which is collimated through a lens and a dichroic mirror. As mentioned above, Cy5 is chosen as the acceptor probe, and the strongest emission bands of the donor probe and acceptor probe are 620 nm and 665 nm, respectively. The transmission band of the dichroic mirror should include the two wavelengths mentioned above. The desired wavelength range of the emitted fluorescence is obtained using filter2 or filter3, which is selected by a filter switching component to be used in turn. The center wavelengths of filter2 and filter3 are selected by considering the strongest emission bands of the donor probe and acceptor probe. Related parameters of those filters are presented in Table 1.

Finally, to enhance the sensitivity of the measurement, a photon-counting photomultiplier tube (PMT), model CH277 from Beijing Hamamatsu Photon Techniques Inc., Beijing, China. was selected. According to the specifications of the CH277, the pulse resolution is 20 ns; the width of the output pulse is 10 ns; the maximum linear counting rate is 5 × 10^6^ s^−1^; and the typical dark count rate is 150 s^−1^.

A photograph of the photon-counting PMT (model CH277) is shown in Figure 4.

The wireframe diagram of the filter switching component is illustrated in Figure 5. Filter2 and filter3, as shown in Figure 3, are fixed onto a fan-shaped structural component. The actuator for the filter switching component is a stepper motor, which drives the rotation of the fan-shaped structural component using a synchronous belt. The fan-shaped structural component is coaxially equipped with positioning plates to facilitate system positioning via a photoelectric sensor.

In the design phase, it is ensured that when the photoelectric sensor is obstructed, one of the filters assumes its operational position. The positioning of the other filter is determined by several factors: the angular relationship of the two filters relative to the fan-shaped structural component’s rotational axis, the transmission ratio of the switching mechanism, and the step angle of the motor. The stepper motor 14H2027-050-4AL-09-J from Jiangsu DINGS’ Intelligent Control Technology Co. Ltd., Changzhou, Jiangsu, China, featuring a step angle of 1.8° and a rated phase current of 0.5A, has been selected for its suitability. The motion control resolution is further refined through the subdivision settings of the driver chip, enhancing the precision of the system’s operations.

The timing diagram for a single measurement is depicted in Figure 6.

In this diagram, *T*_1_ represents the delayed time, and *T*_2_ is the integration time. Once the specified center wavelength filter is in position, the xenon flash lamp is triggered to emit light. The triggering signal serves as the synchronous signal, initiating the delayed timer. When the specified delayed time *T*_1_ elapses, the PMT signal is collected, and the integration timer is synchronized to begin counting. After the integration time *T*_2_ concludes, PMT data collection stops, achieving a single measurement. In practical applications, multiple measurements are often taken to calculate the mean value, enhancing measuring result accuracy.

The structure diagram of the control system of the measuring module is shown in Figure 7.

In consideration of cost-effectiveness and peripheral versatility, the STM32F103VCT6 microcontroller has been selected as the primary control chip for the measuring module control system, as depicted in Figure 8. The main controller chip regulates the output voltage of a control amplifier using a digital potentiometer, thus adjusting the output voltage of the high-voltage power supply for the xenon lamp, ultimately determining the xenon lamp’s optical power output. The main controller issues pulse and direction control signals, which drive the filter-switching motor through the stepper motor driver. Initially, the filters are positioned using optocouplers, which serve as feedback signals. The photon-counting PMT processing module is designed as a high-frequency, high-precision counting module, comprising an on-chip 8-bit counter and an external 8-bit counter, SN74F161ADR. Compared to off-the-shelf alternatives, this custom counting module offers cost advantages. Once the filters are correctly positioned, the main controller triggers the high-voltage power supply of the xenon lamp, causing it to emit light. Following a predetermined delay period, the main controller synchronously activates both the on-chip 8-bit counter and the external 8-bit counter until the integration time is completed. Communication between the main controller chip and the instrument is facilitated through an RS232 interface.

The A3983SLPTR-T is capable of driving bipolar stepper motors in full-step, half-step, 1/4-step, and 1/8-step modes of operation. When supplied with a voltage of 35 V, it can drive currents of up to ±2 A. The subdivision settings are determined by the logic levels on the MS1 and MS2 pins. In contrast, the output current is determined by the voltage on the REF pin and the resistors connected to the SENSE1 and SENSE2 pins, as illustrated in Figure 8. The truth table for subdivision settings is presented in Table 2.

From the truth table, it is evident that the subdivision setting configured in Figure 8 corresponds to an eight-step subdivision.

The formula for calculating the driving current is as follows:(1)$$I_{TripMAX}=\frac{V_{REF}}{8\times R_{47}}=\frac{3.3\times R_{46}}{8\times \left(R_{45}+R_{46}\right)\times R_{47}}=\frac{3.3\times 10}{8\times \left(6.49+10\right)\times 0.5}=0.5\left(A\right)$$

The physical representation of the electronic control board for the measuring module is depicted in Figure 9.

6061 T651 aluminum alloy was used for construction, and black anodizing treatment was produced to meet the application requirements of optical components. The physical image of the measuring module is shown in Figure 10.

## 4. Instrument Design

Based on the methodological reaction process and following a modular design approach, the main components of the instrument include a pipetting module, tip immobilized probe (TIP) and dilution cup module, reagent and sample module, detection module, electronic control module, and waste container. The overall layout is shown in Figure 11. Reagents and samples are aspirated by using TIP to avoid carryover. The liquid handling module extracts the TIP from the TIP and dilution cup module and then transfers it to a specified position in the reagent and sample module to aspirate reagents or samples. After that, the TIP is moved to the sample port in the detection module for filling and finally disposed to the waste container.

The detection module includes a temperature control system to provide a stable ambient temperature for the reaction. After a specified incubation time for the immune complex formed by reagents and samples, the detection module transports the 96-well plate to the specified detection port while closing the sample port simultaneously. The detection module carries out the measurements.

Due to the high sensitivity of the photon-counting PMT, it should be used in a darkroom to reduce measuring noise caused by parasitic light and to avoid the possibility of permanent damage caused by strong light. A structural diagram of the linkage mechanism between the detection port and the sample port is shown in Figure 12.

The linkage slider is attached to the light-shielding box and slides in the direction of the line connecting the sample port and the detection port. The linkage slider has holes at both ends, one of which is a sample through-hole and the other a detection through-hole. The linkage motor, connected via an eccentric component, drives the linkage skateboard to move on the light-shielding box, allowing that sample through-hole to reach the position of the sample port or detection through-hole to reach the position of the detection port.

The linkage mechanism between the detection port and the sample port ensures that when the sample port is closed, the detection port is open, and vice versa. This mechanism can prevent light leakage during sampling and detection. To enhance light shielding, the detection port and sample port are designed with corresponding convex and concave structures, and an integrated light shield is installed on the top plate.

The assembled and debugged instrument is depicted in Figure 13.

The procedure for testing a single sample is as follows:(a)Begin by adding two reagents into the reaction vessel using the pipetting module.(b)Introduce the sample into the reaction vessel and perform mixing by repeatedly aspirating and dispensing with the pipetting module.(c)Transport the reaction vessel to the detection module for incubation, with the incubation time pre-set.(d)Transfer the reaction vessel to the measurement position.(e)Send a trigger signal from the controller to the xenon lamp, causing it to emit light and initiating a single measurement cycle.(f)Repeat step (e) for a specified number of cycles.(g)Send control signals from the controller to the stepper motor driver, positioning the 665 nm filter at the measurement position.(h)Send another trigger signal from the controller to the xenon lamp, initiating another single measurement cycle.(i)Repeat step (h) for a specified number of cycles.(j)Send control signals from the controller to the stepper motor driver, positioning the 620 nm filter at the measurement position.

## 5. Reagent Preparation

### 5.1. Reagents and Chemicals

Phosphate-buffered saline (PBS) and bovine serum albumin (BSA) were purchased from Titan Scientific Co., Ltd. (Shanghai, China), and dimethyl sulfoxide (DMSO) was purchased from Solarbio Life Science (Beijing, China). DiSulfo-Cy5-NHS ester was purchased from Ruixi Biotech Co., Ltd. (Xi’an, China). ProClin 300 was obtained from Sigma-Aldrich (Burlington, MA, USA). Europium (Eu^3+^) cryptate NHS ester was a gift from Prof. Liu at Suzhou Institute of Biomedical Engineering and Technology, Chinese Academy of Sciences. All antibodies used for reagent preparation were obtained from commercial sources.

### 5.2. Construction of Europium (Eu^3+^) Cryptate and Disulfo-Cy5 Labeled Antibodies

Antibodies (100 μg, 1 mg/mL in PBS, pH 7.2~7.4) were mixed with either europium (Eu^3+^) cryptate NHS ester (1.34 μL of 2 mM stock solution in DMSO) or diSulfo-Cy5-NHS ester (2 μL of 2 mM stock solution in DMSO) and incubated lucifugally at 37 °C for 2 h. The crude products were then purified by size exclusion columns (7 kDa 2 mL, Thermo Fisher Scientific Inc.,Waltham, MA, USA) to afford europium (Eu^3+^) cryptate and disulfo-Cy5 labeled antibodies.

### 5.3. Antigen Detection

Antigen-detecting reagents were prepared with labeled antibodies. Briefly, europium (Eu^3+^) cryptate and disulfo-Cy5 labeled antibodies were diluted in buffer A (0.3% BSA and 0.1% ProClin 300 in PBS, pH 7.2~7.4). To set up the assays, 70 μL of antigen samples was added to 100 μL reagent mix and incubated at 37 °C for 18 min. FRET donor and acceptor signals were measured at 620 nm and 665 nm, respectively.

## 6. Experiment

As a self-designed module, the counting accuracy and precision of the high-frequency, high-precision counting module have a direct impact on the measurement results. To assess its performance, various frequency pulse signals generated by a signal generator, specifically the AFG31052 from Tektronix Inc., were employed to simulate the photon-counting PMT and evaluate the counting module’s coefficient of variation (CV). With an integration time set to 250 μs, the CV values obtained for the counting module are summarized in Table 3. It is evident that all CV values are consistently below 0.01%, thereby satisfying the system’s application requirements.

To test instrument noise, perform the testing by using a blank sample after the instrument has been turned on and reaches a stable operating state. This involves conducting 20 consecutive tests and recording the luminescence values at the 620 nm and 665 nm wavelengths for each test. The arithmetic mean of the luminescence values for each wavelength is calculated, and the instrument noise is determined by using the following formula:(2)$$I_{B}=\bar{I}+2s$$

In the formula, *I_B_* represents instrument noise; $\bar{I}$ is the arithmetic mean of the luminescence values; *s* is the standard deviation.

Test results show that the instrument noise at the 620 nm and 665 nm wavelengths is 41.44 and 10.59, respectively.

Ensuring the accuracy and stability of the temperature in the detection darkroom is critical to maintaining the stability of the luminescence of the immune complexes. To test the temperature-related parameters of the darkroom, a temperature probe with a resolution of no less than 0.1 °C is placed at the specified position in the darkroom. After the temperature display stabilizes, temperature values are measured every 30 s for a duration of 10 min. The mean of the temperature measuring values is compared to the set value to determine the temperature bias B, and half of the difference between the maximum and minimum values represents the temperature fluctuation. The results indicate that when the set temperature is 37 °C, the temperature bias B is 0.06 °C, and the temperature fluctuation is 0.10 °C.

To validate the instrument’s pipetting accuracy and repeatability, the pipetting volumes are set to 10 μL, 20 μL, 50 μL, and 100 μL, respectively, and the weighing method is used. The weighing procedure is as follows:(a)The instrument, degassed distilled water, and other materials are placed in a temperature and humidity-controlled laboratory for several hours to achieve equilibrium.(b)Using a single-cup container, the mass is measured with an electronic balance calibrated to a resolution of 0.01 mg, and the balance reading is zeroed.(c)The aforementioned single-cup container is positioned in the instrument’s testing position, and a specified volume of degassed distilled water is dispensed into the container using a controlled aspiration needle. The mass of the water (*m_a1_*) is then determined using the electronic balance, representing the actual added mass for this trial. Subsequently, the balance reading is reset to zero.(d)Repeat step (c) for a total of 20 sample additions.(e)The actual added volume (*v_i_*) for each trial is calculated by dividing the actual added mass (*m_ai_*) for each trial (*i* = 1~20) by the density of distilled water at the current temperature, and the average value is computed.(f)The coefficient of variation (*CV*) and bias (*B*) for the 20 measurements are calculated according to the following formula:
(3)$$CV=\frac{\sqrt{\frac{\sum_{i=1}^{20}{\left(v_{i}-\bar{v}\right)}^{2}}{20}}}{\bar{v}}\times 100\%$$
(4)$$B=\frac{\bar{v}-v_{T}}{v_{T}}\times 100\%$$

In the formula, $\bar{v}$ is the measured average value of the added sample; *v_T_* is the specified added amount.

The results are presented in Table 4.

Using the luminophore method [38], high-intensity fluorescent molecules were proportionally diluted into six concentrations, covering the concentration range of 0.01 nM to 10 nM. Each sample was measured three times, and the luminescence values at the 620 nm wavelength were recorded. The mean of the three measurements for each sample was calculated. Linear regression was performed with the dilution ratio as the independent variable and the mean measuring results as the dependent variable. The linear regression correlation coefficient is calculated.

The results are shown in Figure 14. It can be observed that within the concentration range of 0.01 nM to 10 nM, the linear regression correlation coefficient is greater than 0.999, indicating excellent linearity.

First, using PCT as an example, standard samples with nine concentrations ranging from 0.02 ng/mL to 50 ng/mL were selected for testing with self-developed reagents. The concentrations included 0.02 ng/mL, 0.05 ng/mL, 0.08 ng/mL, 0.1 ng/mL, 0.2 ng/mL, 0.5 ng/mL, 1 ng/mL, 2 ng/mL, 5 ng/mL, 10 ng/mL, 20 ng/mL, and 50 ng/mL. The results of these tests, along with the limit of quantification (LOQ) at 0.096 ng/mL, are presented in Figure 15.

Similarly, CRP and IL-6 were tested using the same method, with CRP concentrations ranging from 0 ng/mL to 500 ng/mL and IL-6 concentrations ranging from 0 ng/mL to 20 ng/mL. The results, shown in Figure 16, demonstrate excellent linear characteristics. The LOQ for CRP and IL-6 were determined to be 2.70 ng/mL and 2.82 ng/mL, respectively.

The performance comparison between the self-developed instrument and that from Thermo Fisher Scientific is shown in Table 5.

## 7. Conclusions

The TR-FRET immunoassay method combines the FRET technology of reagents with the instrument’s time-resolved measuring technology to enhance specificity and sensitivity. One system based on TR-FRET technology is designed. In the system, a xenon flash lamp is selected as the light source, and a PMT is selected as the detector. In the detection optical path, a filter-switching mechanism is employed to temporally allow the entry of donor fluorescence and acceptor fluorescence into the PMT. The measuring procedure involves first measuring the donor fluorescence. When the xenon lamp is triggered to emit light, a delayed timer is initiated. After the delayed time elapses, PMT signals are collected until the integration time ends. This measurement is repeated several times. Subsequently, the filter is switched to allow the acceptor fluorescence to enter the PMT, and the above-mentioned measuring process is repeated. Thus, the measurement for a sample is completed. Reagents and samples are aspirated by using TIP to avoid carryover. Furthermore, a mechanism that couples the detection port and sample port is designed to ensure no light leakage during sampling or detection.

Experimental results are given as follows. The CV value of the self-designed counting module is better than 0.01%, and the instrument noises for 620 nm and 665 nm are 41.44 and 10.59, respectively. When set at 37 °C, the temperature bias (B) is 0.06 °C, and the temperature fluctuation is 0.10 °C. For sample volumes of 10 μL, 20 μL, 50 μL, and 100 μL, the coefficient of variation (CV) and bias (B) are as follows: 2.56%, 1.86%, 0.39%, −0.01%, and 4.91%, 1.30%, 0.79%, 0.43%. Within the concentration range of 0.01 nM to 10nM, the luminescence values exhibit linear regression correlation coefficients greater than 0.999. For PCT, within the concentration range of 0.02 ng/mL to 50 ng/mL, the linear regression correlation coefficient is greater than 0.999, and the limit of quantification is 0.096 ng/mL. For CRP and IL-6, the testing concentration ranges are 0 ng/mL to 500 ng/mL and 0ng/mL to 20ng/mL, and the limits of quantification are 2.70 ng/mL and 2.82 ng/mL, respectively.

The above experimental results validate the feasibility of the technology and instrument solution.

## Figures and Tables

**Figure 1 sensors-24-01430-f001:**
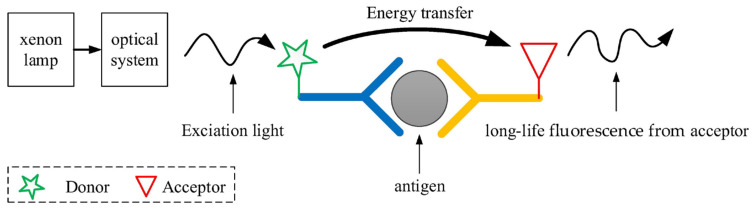
Schematic diagram of fluorescence resonance energy transfer.

**Figure 2 sensors-24-01430-f002:**
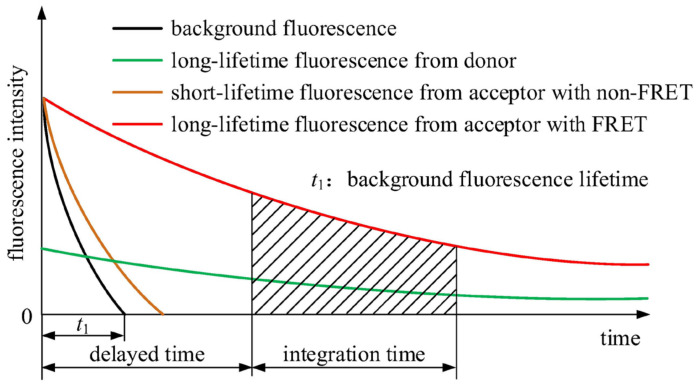
Principle of time-resolved fluorescence resonance energy transfer measurement.

**Figure 3 sensors-24-01430-f003:**
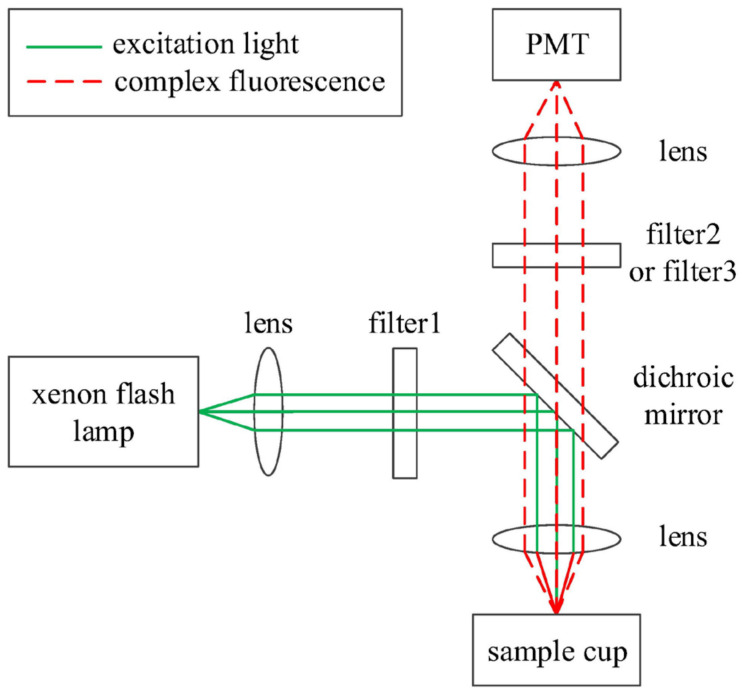
Principle diagram of the optical system of the measuring module.

**Figure 4 sensors-24-01430-f004:**
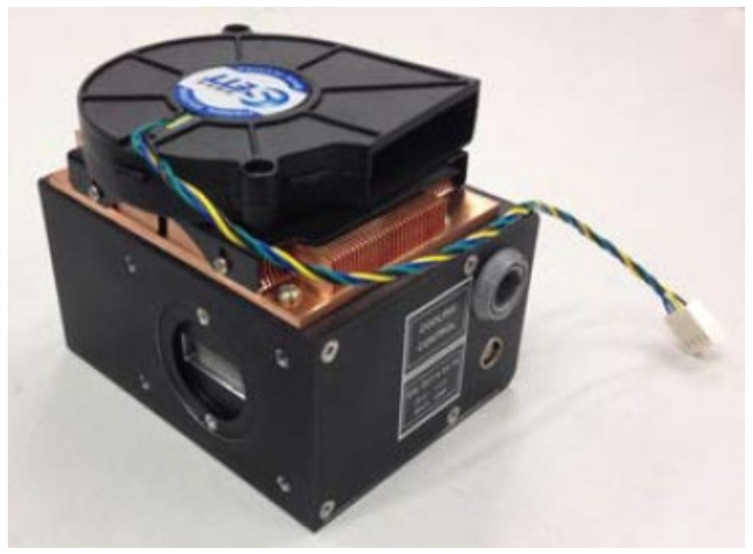
A photograph of the photon-counting PMT CH277.

**Figure 5 sensors-24-01430-f005:**
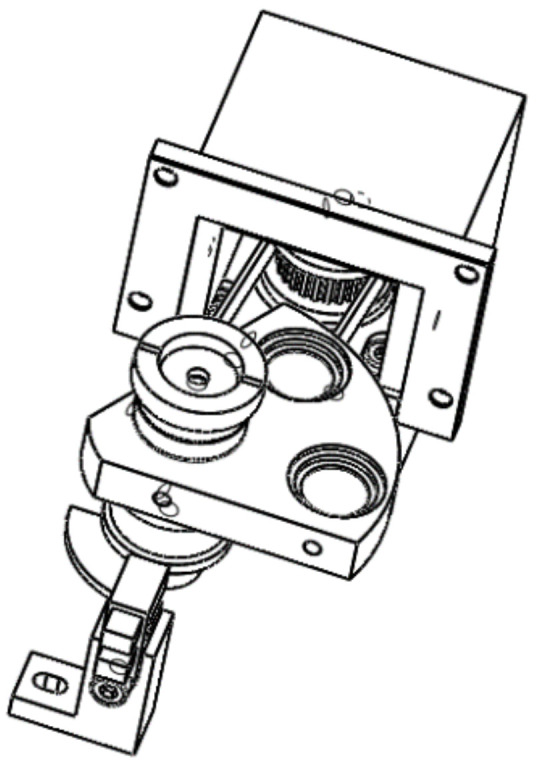
Wireframe of the Filter Switching Module.

**Figure 6 sensors-24-01430-f006:**
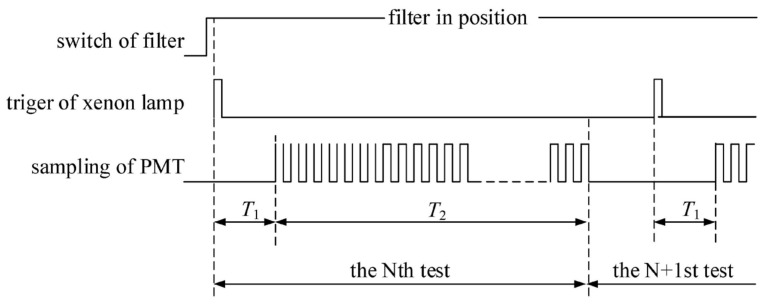
The timing diagram of a single measurement.

**Figure 7 sensors-24-01430-f007:**
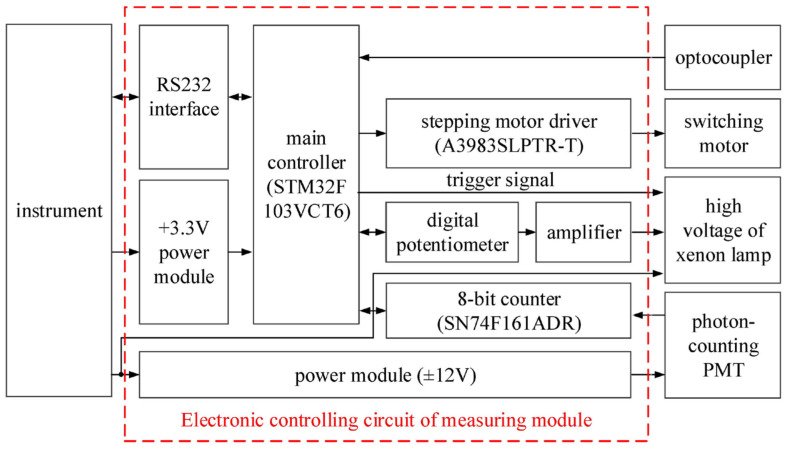
Structure diagram of the control system of the measuring module.

**Figure 8 sensors-24-01430-f008:**
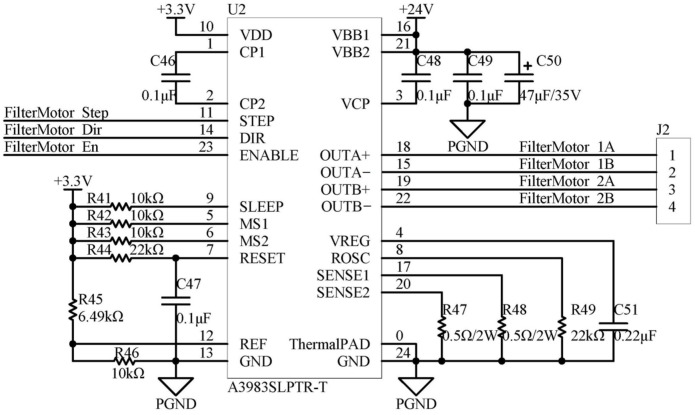
Application circuit of A3983SLPTR-T.

**Figure 9 sensors-24-01430-f009:**
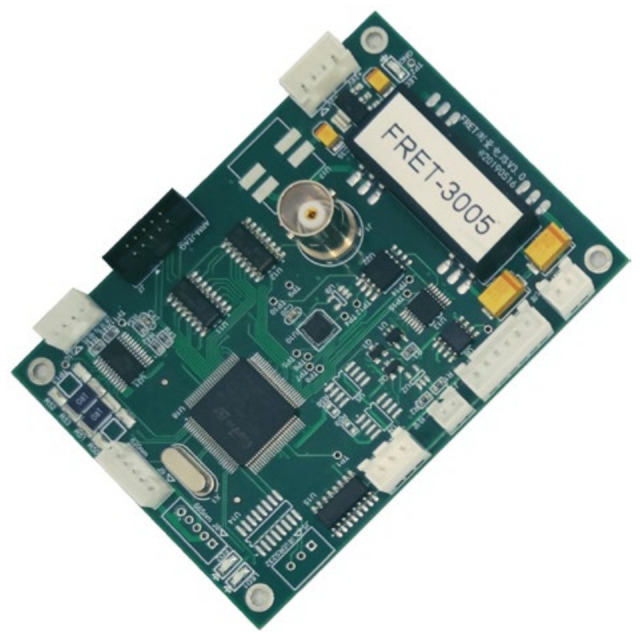
A photograph of the electronic controlling circuit of the measuring module.

**Figure 10 sensors-24-01430-f010:**
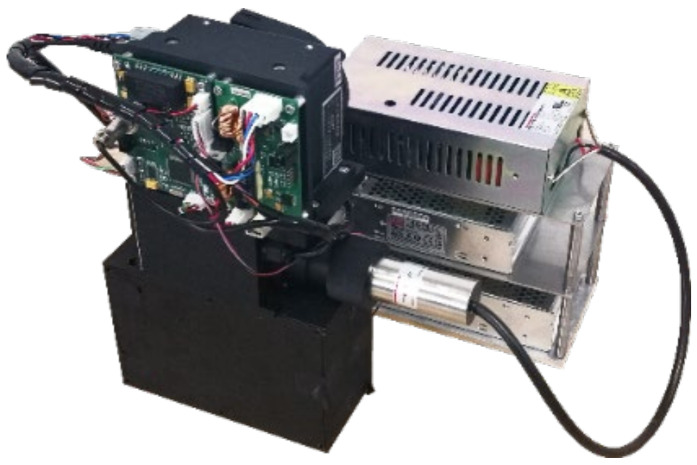
A photograph of the measuring module.

**Figure 11 sensors-24-01430-f011:**
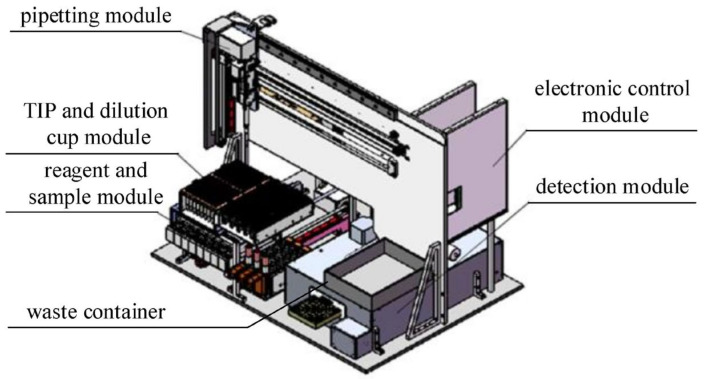
The overall layout of the instrument.

**Figure 12 sensors-24-01430-f012:**
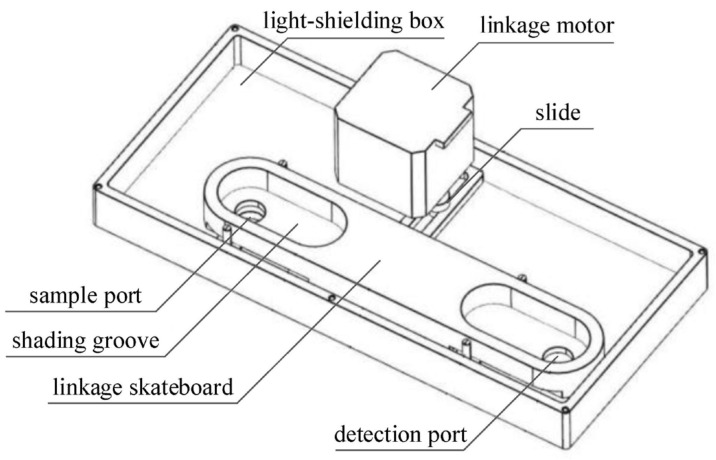
Structural diagram of the mechanical linkage.

**Figure 13 sensors-24-01430-f013:**
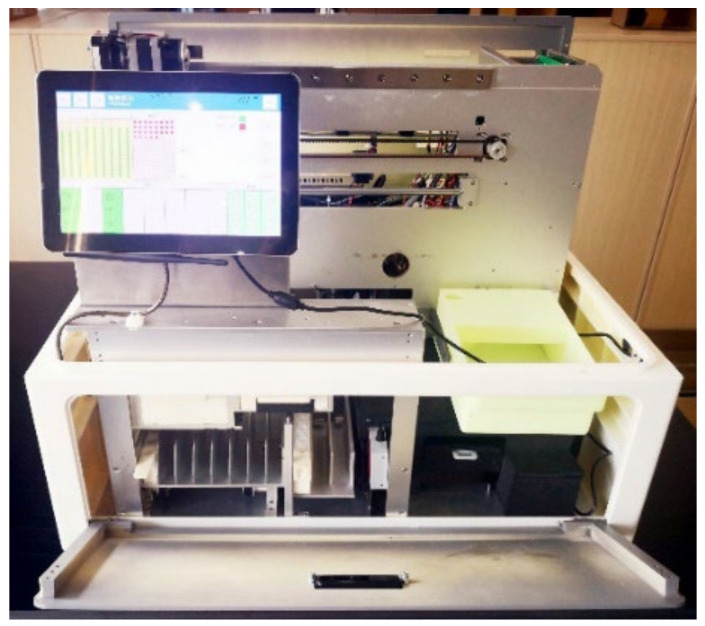
A photograph of the instrument after assembly and calibration.

**Figure 14 sensors-24-01430-f014:**
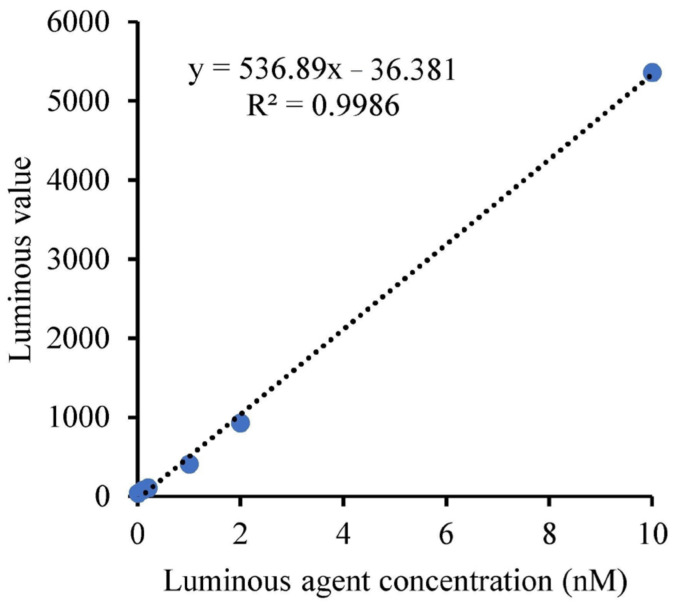
Test curve of luminous linearity of the fluorescent molecule.

**Figure 15 sensors-24-01430-f015:**
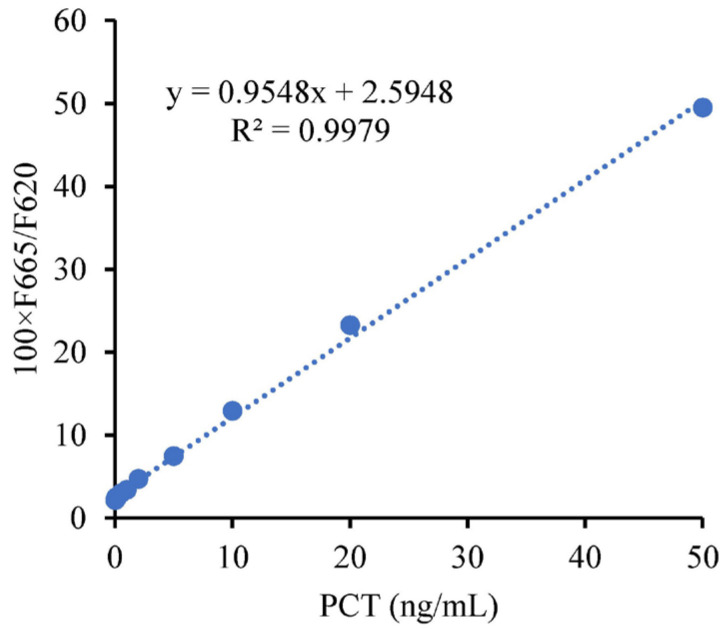
Relationship curve between PCT concentration and measuring results.

**Figure 16 sensors-24-01430-f016:**
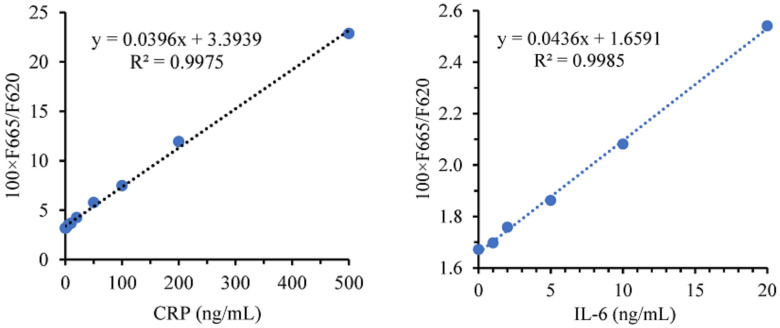
Relationship curve between CRP or IL-6 concentration and measuring results.

**Table 1 sensors-24-01430-t001:** Parameters of the filters used.

Label	Center Wavelength (nm)	Bandwidth (nm)	Optical Density
filter1	320	25	OD6
filter2	620	20	OD6
filter3	665	8.5	OD6

**Table 2 sensors-24-01430-t002:** Micro-step resolution truth table.

MS1	MS2	Micro-Step Resolution
L	L	Full Step
H	L	Half Step
L	H	Quarter Step
H	H	Eighth Step

**Table 3 sensors-24-01430-t003:** CV of the counting module.

Signal Frequency	CV (%)	Signal Frequency	CV (%)
5 MHz	0.0039	30 MHz	0.0042
	0.0000		0.0041
	0.0000		0.0042
	0.0039		0.0043
10 MHz	0.0028	40 MHz	0.0031
	0.0000		0.0032
	0.0020		0.0031
	0.0020		0.0031
20 MHz	0.0024	50 MHz	0.0041
	0.0023		0.0049
	0.0032		0.0050
	0.0023		0.0043

**Table 4 sensors-24-01430-t004:** Test results of accuracy and repeatability of pipetting.

Order Number	Sampling Volume (μL)	CV	B
1	10	2.56%	4.91%
2	20	1.86%	1.30%
3	50	0.39%	0.79%
4	100	−0.01%	0.43%

**Table 5 sensors-24-01430-t005:** Performance comparison between self-developed and other instruments.

Item	Self-Developed Instrument	Instrument from Thermo Fisher Scientific
Corollary reagents	PCT, CRP, IL-6, SAA (Under development)	PCT, CPP
Direct quantitative linear range of PCT	0.02 ng/mL to 50 ng/mL	0.02 ng/mL to 50 ng/mL
Limit of quantification of PCT	0.096 ng/mL	0.076 ng/mL
Cost	low	high

## Data Availability

The original contributions presented in the study are included in the article, further inquiries can be directed to the corresponding authors.

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
