# Peer review of "Immunoassay System Based on the Technology of Time-Resolved Fluorescence Resonance Energy Transfer"

_sensors, 2024, doi:10.3390/s24051430_

Round 1

Reviewer 1 Report

Comments and Suggestions for Authors

Dear Authors,

While I strongly agree that in-vitro diagnostics is an important field where significant gains can be made by introducing assays based on (advanced) fluorescent readout and that in this context your work can be valuable, I do not believe that the paper as it is currently written merits publication.

My biggest comment is that the current paper does not seem to consider the fact that several setups for measuring TR-FRET, such as the one constructed in this paper, already exist and are commercially available.

For this paper to make a meaningful contribution to the field, the newly developed set-up should be contrasted to the systems already published and/or on the market. This would involve:

-Restructuring/expanding the introduction (lines 34-64) to express the current market position of TR-FRET-based analysis (quantitatively if possible).

-Adding a brief section (eg. after line 115) where you clearly state the goal of this research (is the set-up cheaper than alternatives?, can it do something that no other set-up can?, is it easy to construct yourself?, is it more sensitive?, ...)

-Enhancing the technical character of section 3 (lines 117-220) since to some extent this section is the main contribution of the work (which specific parts were used? how was the electronic measuring module constructed and programmed? which materials were used for construction? as well as how these choices helped you attain the goal of your research e.g. "the use of off-the-shelf microcontrollers allows for costs to be minimized")

-comparing the performance of the system to existing methods (1-to-1 if possible, otherwise a comparison with published figures of merit of other instruments)

besides this main comment, I also have 5 additional remarks:

- What is the reason for the synthetic test of the counting module (table 1)? Are the obtained results not already evident from the specs of the manufacturer?

-Can you quantify the limit of detection/quantification?

-Can you provide the labels and filters that were used in the measurements?

-Can you explain or provide a reference for the "weighing method" and "lumiphore method" as well as a definition for Bias as used in Table 2

-how is the Procalcitonin assay performed? is it a commercial assay? Why is this a good benchmark?

Author Response

Thank you for your comments, We have carefully considered all comments from the reviewers and revised our manuscript accordingly. According to the reviewers' comments, please refer to the attachment for specific response content.

Reviewer 2 Report

Comments and Suggestions for Authors

In the manuscript “Immunoassay system based on the technology of time-resolved 2 fluorescence resonance energy transfer” by Z. Xu, H. Zhou, L Li, Z. Chen, X. Zhang, Y. Feng, J. Wang, Y. Li and Y. Wu the authors describe a method of in vitro immunodiagnostics based on registration of the time-resolved fluorescence of an indicator, stimulated by Förster ressonance energy transfer (FRET) from the donor of energy with extremely long fluorescence lifetime, such as lanthanide ions. The authors describe the details of the equipment used for this study. The authors demonstrate high sensitivity of this equipment using as an example the novel inflammatory marker Procalcitonin (PCT). The presented method and the respective equipment are interesting and perspective and may attract attention of specialists in immunodiagnostics. The manuscript can be accepted for publication in the journal “Sensors”.

Comments:

1.      While indicating Procalcitonin as an immunodiagnostic fluorescent indicator, the authors for any reason do not indicate the energy donor with a long fluorescence lifetime they used.

2.      The English language of the manuscript must be carefully checked and corrected. Comments on the Quality of English Language The English language of the manuscript must be carefully checked and corrected.

Author Response

(The authors gave the same response as above.)

Reviewer 3 Report

Comments and Suggestions for Authors

In the manuscript “Immunoassay system based on the technology of time-resolved fluorescence energy transfer,” the authors present the application of time-resolved Foster energy transfer (TR-FRET) methods and instrument device applied to the immunoassay.

In my opinion, this work is interesting and appropriate for the Journal but, I recommend the publication after a major revision. In particular: 

A) The introduction is not very clear. In particular:

1. The authors should improve the introduction section with attention to describing in detail some concepts motioned such as long-lifetime, time-resolved, etc.

2. The main aim of the work is not completely clear and not well-presented. Review the from line 60 to line 64.   

B) The figure 1 is not clear. In particular: 

1. What do they mean by light emitted? Maybe they refer to excitation light? Is it correct? 

2. The figure caption should be improved.

Please, the authors provide it.

- The figure 2 is not clear. In particular: 

1. the figure caption should be improved.

Please, the authors provide it.

C why in the instrument design and realization the authors decide to use xenon lamp and not more easily low cast diode laser and/or LED?

D) Figure 13 is not clear. In particular, the figure caption should be improved. Please, the authors provide it.

E) Have the authors evaluated the performance of the instrument on the real immunoassay experiments? if yes the results should be included in the manuscript.

Comments on the Quality of English Language

No comment

Author Response

(The authors gave the same response as above.)

Round 2

Reviewer 1 Report

Comments and Suggestions for Authors

All remarks where resolved.

Would remove or reword lines 146-149, since this discusses editorial remarks?

Comments on the Quality of English Language

The text is perfectly readable, though at times oddly phrased

Author Response

(The authors gave the same response as above.)

Reviewer 3 Report

Comments and Suggestions for Authors

The requested modifications were done from the authors. The manuscript can be accepted for publication.

Author Response

(The authors gave the same response as above.)
